# Steroids from the Deep-Sea-Derived Fungus *Penicillium*
*granulatum* MCCC 3A00475 Induced Apoptosis via Retinoid X Receptor (*RXR*)-α Pathway

**DOI:** 10.3390/md17030178

**Published:** 2019-03-19

**Authors:** Chun-Lan Xie, Duo Zhang, Jin-Mei Xia, Chao-Chao Hu, Ting Lin, Yu-Kun Lin, Guang-Hui Wang, Wen-Jing Tian, Zeng-Peng Li, Xiao-Kun Zhang, Xian-Wen Yang, Hai-Feng Chen

**Affiliations:** 1School of Pharmaceutical Sciences, Xiamen University, South Xiangan Road, Xiamen 361005, China; xiechunlanxx@163.com (C.-L.X.); 32320151154210@stu.xmu.edu.cn (D.Z.); 32320171153261@stu.xmu.edu.cn (C.-C.H.); linting@xmu.edu.cn (T.L.); guanghui@xmu.edu.cn (G.-H.W.); tianwj@xmu.edu.cn (W.-J.T.); 2Key Laboratory of Marine Biogenetic Resources, South China Sea Bio-Resource Exploitation and Utilization Collaborative Innovation Center, Third Institute of Oceanography, Ministry of Natural Resources, 184 Daxue Road, Xiamen 361005, China; xiajinmei@tio.org.cn (J.-M.X.); yukunlin223@163.com (Y.-K.L.); lizengpeng@tio.org.cn (Z.-P.L.)

**Keywords:** marine-derived fungus, *Penicillium granulatum*, ergostanes, anti-tumor, nuclear receptor

## Abstract

Five new ergostanes, penicisteroids D−H (**1**−**5**), were isolated from the liquid culture of the deep-sea-derived fungus *Penicillium granulatum* MCCC 3A00475, along with 27 known compounds. The structures of the new steroids were established mainly on the basis of extensive analysis of 1D and 2D NMR as well as HRESIMS data. Moreover, the absolute configurations of **1** were confirmed unambiguously by the single-crystal X-ray crystallography. Compounds **2** and **4**–**7** showed moderate antiproliferative effects selectively against 12 different cancer cell lines with IC_50_ values of around 5 μM. Compounds **2** and **6**, potent *RXRα* binders with Kd values of 13.8 and 12.9 μM, respectively, could induce apoptosis by a Retinoid X Receptor (*RXR*)-*α*-dependent mechanism by regulating RXRα transcriptional expression and promoting the poly-ADP-ribose polymerase (PARP) cleavage. Moreover, they could inhibit proliferation by cell cycle arrest at the G0/G1 phase.

## 1. Introduction

Retinoid X receptor-α (*RXRα*), one of the most promising members of the nuclear receptor superfamily, plays an important role for the treatment of cancer, metabolic, and neurodegenerative diseases [1,2,3,4]. It can form heterodimers with retinoic acid retinoid (*RAR*), thyroid hormone receptor (*TR*), nerve growth factor-induced gene B (*NGFIB*), etc. [5,6]. As a matter of fact, thirteen percent of all Food and Drug Administration (FDA)-approved drugs are targeting nuclear receptors [7,8]. In 1999, the retinoid X receptor (*RXR*) agonist, targretin^TM^ (bexarotene), was approved by the FDA for the treatment of cutaneous T-cell lymphoma, stimulating scientists around the world to search for more *RXR* transcriptional inhibitors [9,10].

*Penicillium granulatum* MCCC 3A00475 was isolated from a deep-sea sediment sample of the Antarctic Ocean. Previously, the chemical investigation on the rice static fermentation extract of this fungus provided novel spiro-diterpenoids, alkaloids, and steroids [11,12]. Inspired by OSMAC (One-Strain Many Compounds) strategy, the fungus was subjected to further investigation using different fermentation conditions. Interestingly, under the agitated fermentation in liquid medium, its secondary metabolic profile changed dramatically. Moreover, the crude extract showed potent cytotoxicity against several cancer cells. Therefore, a large-scale fermentation was conducted, followed by a systematic chemical isolation. As a result, five new and 27 known compounds were obtained (Figure 1). Compounds **2** and **4**–**7** showed moderate inhibitory effects selectively against 12 different cancer cell lines with IC_50_ values of around 5 μM. The mechanism study indicated they could not only induce apoptosis through an RXRα-dependent pathway but also inhibit the proliferation by cell cycle arrest at G0/G1 phase. Herein, we report the isolation, structure elucidation, and cytotoxicity against a panel of cancer cell lines of these compounds.

## 2. Results and Discussion

Compound **1** was isolated as colorless needle crystals. The sodium adduct molecular ion peak at *m/z* 479.3547 in the HRESIMS (High Resolution Electrospray Ionization Mass Spectroscopy) indicated its molecular formula as C_30_H_48_O_3_, requiring six degrees of unsaturation. The ^1^H NMR spectrum (Appendix A) exhibited four methyl doublets [*δ*_H_ 0.82 (3H, d, *J* = 6.7 Hz, Me-26), 0.84 (3H, d, *J* = 6.7 Hz, Me-27), 0.87 (3H, d, *J* = 6.9 Hz, Me-28), 1.06 (3H, d, *J* = 6.7 Hz, Me-21)], three methyl singlets [*δ*_H_ 0.93 (3H, s, Me-18), 1.03 (3H, s, Me-19), and 1.98 (3H, s, Me-2′)], three olefinic protons [*δ*_H_ 5.32 (1H, d, *J* = 5.0 Hz, H-6), 5.19 (2H, m, H-22 and H-23)], and two oxymethine [*δ*_H_ 3.38 (1H, m, H-3), 5.00 (1H, dt, *J* = 7.6, 3.6 Hz, H-16)]. The ^13^C and DEPT (Distortionless Enhancement by Polarization Transfer) spectra (Appendix A) revealed the presence of 30 carbons, including seven *sp*^3^ methyls, seven methylenes, twelve methines (two oxygenated *sp*^3^ and three *sp*^2^), and four non-protonated carbons (two *sp*^2^ and two *sp*^3^). Altogether, the 1D NMR spectroscopic data (Table 1 and Table 2) indicated one acetoxyl moiety [*δ*_H_ 1.98 (3H, s, Me-2′); *δ*_C_ 172.3 (s, C-1′), 21.7 (q, Me-2′)] and a 28-carbon skeleton. Since four olefinic carbons and one carboxyl moiety accounted for three unsaturation degrees, **1** should be a tetracyclic molecule. The assumption, along with the presence of four methyl doublets and two methyl singlets, suggested a steroid skeleton for **1**.

In the COSY (Correlation Spectroscopy) spectrum (Appendix A), a long chain was constructed according to correlations from H-3 (*δ*_H_ 3.38 m) via H-4 at *δ*_H_ (2.20 m) to the allyl proton H-6 (*δ*_H_ 5.32, d, *J* = 5.0 Hz), and subsequently to H-7 (*δ*_H_ 1.93, ddd, *J* = 12.3, 5.2, 2.2 Hz)/H-8 (*δ*_H_ 1.48 m)/H-14 (*δ*_H_ 1.0 m)/H-15 (*δ*_H_ 2.30 m and 1.02 m)/H-16 (*δ*_H_ 5.00, dt, *J* = 7.6, 3.6 Hz)/H-17 (*δ*_H_ 1.29 m)/H-20 (*δ*_H_ 2.53 m)/H-22 (*δ*_H_ 5.19 m), from H-3 via H-2 (*δ*_H_ 1.45 m) to H-1 (*δ*_H_ 1.08 dd, *J* = 13.2, 3.9 Hz), from H-8 via H-9 (*δ*_H_ 0.96 m) to H-11 (*δ*_H_ 1.57 m)/H-12 (*δ*_H_ 2.04, dt, *J* = 12.5, 3.3 Hz; 1.28 m), and from H-20 to Me-21 (*δ*_H_ 1.06, d, *J* = 6.7 Hz). In addition, another fragment could be deduced by the COSY correlations from two methyl doublets of Me-26 and Me-27 via H-25 (*δ*_H_ 1.40 m) to H-24 (*δ*_H_ 1.76 m), while H-24 to H-23 (*δ*_H_ 5.19 m) and Me-28 (Figure 2). Since the chemical shifts of the two *sp*^2^ methine protons were overlapped at *δ*_H_ 5.19 (m), the above deduced two fragmentations could not be connected according to the COSY spectrum. However, it could be easily resolved by the HMBC (Heteronuclear Multiple-bond Correlation) cross peaks of H-20, H-24, Me-21, and Me-28 (Appendix A). Furthermore, the HMBC correlations of two methyls of Me-18 and Me-19 constructed four rings of the steroid skeleton. In addition, the connection of the acetoxyl moiety to C-16 was proved by the HMBC correlation of H-16 to the carboxyl carbon at *δ*_C_ 172.3. On the basis of the above evidence, the planar structure of **1** was then assigned as 16-acetoxy-3-hydroxyergost-5,22*E*-diene, the 7-dehydroxy derivate of penicisteroid A [13].

The relative configuration of **1** was established according to the coupling constants and NOESY (Nuclear Overhauser Effect) spectrum (Appendix A). The chair conformations of rings A and B and a trans-relationship between them were established by the large half-peak-width of H-3 (W1/2 = 15.9 Hz) as well as the NOESY correlations of Me-19 to H-1a (*δ*_H_ 1.86)/H-2a (*δ*_H_ 1.45)/H-8 (*δ*_H_ 1.48), Me-18 to H-8/H-12a (*δ*_H_ 2.04), while H-3 to H-1b (*δ*_H_ 1.08)/H-2b (*δ*_H_ 1.77) (Figure 2). However, the terrible overlapped signals of H-9 (*δ*_H_ 0.96 m)/H-14 (*δ*_H_ 1.00 m)/H-15a (*δ*_H_ 1.02 m) and H-12a (*δ*_H_ 1.28 m)/H-17 (*δ*_H_ 1.29 m) make it impossible to establish its relative configuration. Fortunately, an orthorhombic crystal was obtained from MeOH. By the single X-ray diffraction analysis using Cu–Kα radiation(Appendix A), the absolute configurations of C-3 and C-16 were both assigned as *S* (Figure 3). Accordingly, **1** was unambiguously established as 16*β*-acetoxy-3*β*-hydroxyergost-5,22*E*-diene and named penicisteroid D.

Compound **2** was isolated as an amorporous powder. The molecular formula, C_30_H_48_O_5_, was assigned by the sodium adduct molecular ion peak at *m/z* 511.3383 in the HRESIMS, indicating six degrees of unsaturation. The ^1^H and ^13^C NMR spectra (Appendix A) exhibited 30 carbons, including four doublet and three singlet methyls, five methylenes, 14 methines (four oxygenated and three olefinic), and four non-protonated carbons (one olefinic and one carboxyls). These signals were closely similar to those of **1** except for two additional hydroxyls in **2**. In the COSY spectrum (Appendix A), correlations were found of H-8 via the oxymethine proton at *δ*_H_ 3.75 to H-6 and via H-9 to another oxymethine proton at *δ*_H_ 4.27, suggesting connections of hydroxyls at C-7 and C-11 positions, respectively. Further confirmation was found by the HMBC correlations of H-7 to C-5/C-6/C-8/C-9 and H-11 to C-8/C-9/C-10 (Appendix A). The large coupling constant of *J*_H7,H8_ (*J* = 7.3 Hz) was indicative of its axial α-orientation and the small coupling constants of *J*_H9,H11;H11,H12_ (*J* = 3.6, 2.9 Hz) pointed to its axial-orientation. This was confirmed by the NOESY correlations of H-9 to H-7/H-11, H-7 to H-14, and H-8 to Me-18/Me-19 (Appendix A). Accordingly, **2** was established as 16*β*-acetoxy-3*β*,7*β*,11*β*-trihydroxyergost-5,22-diene and named penicisteroid E.

Compound **3** had a molecule formula of C_30_H_50_O_7_ as deduced from the sodium adduct molecular ion at *m/z* 545.3454 [M+Na]^+^ in the HRESIMS. Its ^1^H and ^13^C NMR spectroscopic data (Appendix A) greatly resembled those of penicisteroid C [14], expect that an additional hydroxy group was found at the C-5 position. The assumption was confirmed by the HMBC correlations (Appendix A) of Me-19 to C-5 at *δ*_C_ 77.9 and by the COSY cross peaks (Appendix A) of H-6 (*δ*_H_ 3.37 d, *J* = 4.5 Hz) via H-7 (*δ*_H_ 3.68 dd, *J* = 9.8, 4.6 Hz) to H-8 (*δ*_H_ 1.94 m). By detailed analysis of the HSQC, COSY, HMBC, and NOESY (Appendix A) spectra, **3** was then determined as 16*β*-acetoxy-3*β*,5*α*,6*β*,7*β,*11*β*-pentahydroxyergost-22*E*-ene and named penicisteroid F. 

The molecular formula of **4** was determined as C_30_H_50_O_6_ based on the sodium adduct ion peak in its HRESIMS spectrum. The ^1^H and ^13^C NMR spectroscopic data (Appendix A) of **4** were close similar to those of **3** except that the oxygenated methine (*δ*_C_ 68.0) at the C-11 position in **3** was replaced by the methylene (*δ*_C_ 22.2) in **4**, suggesting the absence of the hydroxyl group at C-11. This assumption was evidenced by the significant upshift of C-12 from *δ*_C_ 50.2 to *δ*_C_ 41.3 and C-9 from *δ*_C_ 48.4 to *δ*_C_ 45.4. Further confirmation was obtained by the COSY correlations (Appendix A) of H-12 (*δ*_H_ 2.01 dt, *J* = 12.7, 3.1 Hz) to H-11 (*δ*_H_ 1.42 m). Accordingly, **4** was defined as 16*β*-acetoxy-3*β*,5*α*,6*β*,7*β*-tetrahydroxyergost-22*E*-ene, and named penicisteroid G.

Compound **5** was isolated as a white powder. The molecular formula of C_30_H_50_O_5_ was assigned according to the sodium adduct ion peak at *m/z* 513.3570 in its HRESIMS spectrum. Comparison of the ^1^H and ^13^C NMR spectra (Appendix A) of **5** and **4** showed they were very similar except that the oxygenated non-protonated carbon (*δ*_C_ 77.8) at the C-5 position in **4** was replaced by a methine moiety (*δ*_C_ 47.4) in **5**. This was evidenced by the HMBC correlations (Appendix A) from H-19 (δ_H_ 1.06) to C-1 (δ_C_ 39.7), C-5 (δ_C_ 47.4), C-9 (δ_C_ 53.9), and C-10 (δ_C_ 35.9). Thus, **5** was established as 16*β*-acetoxy-3*β*,6*β*,7*β*-trihydroxyergost-22*E*-ene, and named penicisteroid H.

By comparison of the NMR and MS data with those published in the literature, 27 known compounds were determined to be penicisteroid A (**6**) [13], penicisteroid C (**7**) [14], anicequol (**8**) [15], ergosta-7,22-diene-3*β*,5*α*,6*β*,9*α*-tetraol (**9**) [16], (22*E*,24*R*)-3*β*,5*α*-trihydroxy-ergost-7,22-dien-6-one (**10**) [17], (3*β*,5*α*,6*β*,22*E*)-ergosta-7,22-diene-3,5,6-triol (**11**) [18], (3*β*,5*α*,6*β*,22*E*)-6-methoxyergosta-7,22-diene-3,5-diol (**12**) [18], 5*α*,6*α*,8*α*,9*α*-diepoxy-(22*E*,24*R*)-ergoxt-22-ene-3*β*,7*β*-diol (**13**) [19], ergosterol peroxide (**14**) [20], ergosterol (**15**) [21], topsentisterol D3 (**16**) [22], (24*S*)-24-ethylcholesta-3*β*,5*α*-diol-6-one (**17**) [23], (24*S*)-24-ethylcholesta-3*β*,5*α*,6*α*-triol (**18**) [23], incisterol A2 (1**9**) [24], conidiogenones B (**20**) [25], conidiogenone G (**21**) [25], conidiogenone D (**22**) [26], conidiogenone C (**23**) [26], conidiogenones I (**24**) [26], meleagrin (**25**) [27], roquefortine C (**26**) [12], roquefortine F (**27**) [28], (5*S*)-5-(1H-indol-3-ylmethyl)-2,4-imidazolidione (2**8**), sorbicillin (**29**) [29], 2′,3′-dihydrosorbicillin (**30**) [30], trichodimerol (**31**) [31], and dihydrotrichodimerol (**32**) [31].

All 19 steroids (**1**–**19**) were tested for cytotoxicity against 12 cancer cell lines of SHG-44, HepG2, A549, BIU-87, BEL-7402, ECA-109, Hela-S3, PANC-1, SW620, HcT116, MCF-7, and MB-231. They showed selectively cytotoxicity against A549, BIU-87, BEL-7402, ECA-109, Hela-S3, and PANC-1 cells. However, none exhibited antiproliferative effect against SW620, HcT116, MCF-7, and MB-231 cancer cell lines (Table 3). Further investigation by flow cytometry (Figure 4) and the Western blotting (Figure 5) indicated compounds **2** and **4**–**7** could induce apoptosis in A549 cells. Moreover, compounds **2** and **6** could also inhibit cell proliferation by cell cycle arresting at G0/G1 phase (Figure 6).

To further investigate the apoptosis mechanisms, **2** and **6** were tested for transcriptional activities on two nuclear receptors, *RXRα* and *Nur77* (also called *NGFIB*), using the dual-luciferase reporter gene assay. However, they did not display positive effects on *Nur77*. Instead, they could significantly decrease the transcriptional activation of *RXRα* induced by 9-cis (Figure 7) in a dose-dependent manner (Figure 8). It indicated that **2** and **6** might have selectivity effected transcriptional activation of nuclear receptors and only acted on *RXRα*.

Furthermore, we used fluorescence quenching assay (Figure 9) to analyze compounds for binding to *RXRα*-LBD and identified compounds **2** and **6** as a potent binder with a Kd of 13.8 μM and 12.9 μM, and the role of *RXRα* was illustrated by data showing that transfection of *RXRα* small interfering RNA (si-RNA), which inhibited *RXRα* expression, abrogated the apoptosis effect induced by compounds **2** and **6** (Figure 10). The above findings indicated that these compounds might have a useful impact on *RXRα*-mediated growth inhibition and apoptosis induction in cancer cells as well as *RXRα*-dependent regulation of gene expression.

## 3. Materials and Methods

### 3.1. General Experimental Procedures

Optical rotations were recorded on an MCP 500 automatic polarimeter (Anton Paar Trading Co. Ltd., Shanghai, China) under 20 °C. Ultraviolet spectra were detected by a UV8000 UV/Vis spectrophotometer (Shanghai Metash instrument Co., Ltd., Shanghai, China). The HRESIMS spectra were measured by a Xevo G2 Q-TOF mass spectrometer (Waters Corporation, Milford, MA, USA). The NMR spectra were recorded on a 400 MHz spectrometer (Bruker, Fällanden, Switzerland) using TMS as the internal standard. Reversed-phase HPLC was performed on a 1260 infinity instrument (Agilent Technologies, San Diego, CA, USA) equipped with the DAD detector. Purifications by column chromatography (CC) were performed on silica gel, Sephadex LH-20, and ODS. The TLC plates were visualized under UV light or by spraying with 10% H_2_SO_4_.

### 3.2. Fungal Identification, Fermentation, and Extract

The fungus of *Penicillium granulatum* MCCC 3A00475 was provided by the Marine Culture Collection of China (MCCC). The isolation and identification were described previously [11,12].

Erlenmeyer flasks (250 mL) containing 100 mL fermentation media were directly inoculated with a quarter plate of the strain spores. After 2 days of incubation at 28 °C on a rotary shaker at 180 r/min, 20 mL seed cultures were transferred into a total of 100 flasks (1 L) containing 380 mL of defined medium (10 g glucose, 20 mL mannitol, 5 g potato peptone, 5 g monosodium glutamate, 3 g yeast extract, 3 g maltose dissolved in 1 L of water, pH 7.5). The flasks were cultured with shaking at 180 rpm and 28 °C for 11 days.

The culture (40 L) was centrifuged to separate the broth and mycelia. The mycelia were exhaustively extracted with EtOAc (ethyl acetate) three times to yield a dark brown gum (30.0 g).

### 3.3. Isolation and Purification

The EtOAc extract was subjected to CC on silica gel by gradient CHCl_3_–MeOH (0→100%) as solvents to give eight fractions (Fr.1–Fr.8). Fraction Fr.1 (662.1 mg) was CC over silica gel using petroleum ether (PE)–EtOAc (10:1) to provide **29** (5.7 mg). Fr.2 (694.3 mg) was separated by CC over Sephadex LH-20 (MeOH), followed by purification using prep-HPLC (20→100% MeOH, 20 mm × 25 cm, 10 mL/min) and CC over silica gel (PE-EtOAc, 10:1) to give **1** (4.5 mg), **14** (89.5 mg), **15** (24.6 mg), **18** (1.0 mg), **20** (11.4 mg), **21** (1.4 mg), **22** (4.9 mg), **23** (3.9 mg), and **24** (1.5 mg). Fr.3 (259.6 mg) was CC over Sephadex LH-20 (MeOH). Further purification by prep-HPLC to give **12** (2.5 mg), **16** (1.5 mg), **31** (6.5 mg), and **32** (10.6 mg). Fr.4 (512.2 mg) was separated by CC on Sephadex LH-20 (MeOH), and then by prep-HPLC (40→100% MeOH, 20 mm × 25 cm, 10 mL/min) to afford **8** (9.1 mg) and **25** (134.2 mg). Fr.5 (109.5 mg) was purified by CC on Sephadex LH-20 (MeOH) to obtain **7** (81.9 mg) and **19** (0.5 mg). Fr.6 (586.3 mg) was subjected to CC on Sephadex LH-20 (MeOH) and prep-HPLC (20→80% MeOH, 20 mm × 25 cm, 10 mL/min) to give **2** (12.5 mg), **10** (5.8 mg), **13** (1.4 mg), **26** (4.3 mg), and **27** (5.0 mg). Fr.7 was separated into three sub-fractions (Fr.7.1–Fr.7.3) by CC on Sephadex LH-20 (MeOH). Fr.7.1 (61.5 mg) was purified by preparative HPLC (20→70% MeOH, 10 mm × 25 cm, 5 mL/min) to afford **9** (0.6 mg), **11** (1.8 mg), and **30** (9.7 mg). Fr.7.2 (168.3 mg) was purified by prep-HPLC (30→80% MeOH, 20 mm × 25 cm, 10 mL/min) to provide **5** (1.6 mg) and **6** (58.5 mg). Fr.7.2 (168.3 mg) was purified by prep-HPLC (65% MeOH, 20 mm × 25 cm, 10 mL/min) to give **17** (2.3 mg) and **28** (1.3 mg). Fr.8 (812.4 mg) was CC on Sephadex LH-20 (MeOH) and preparative HPLC (60→100% MeOH, 10 mm × 25 cm, 5 mL/min) to give compound **3** (3.8 mg) and **4** (3.9 mg).

Penicisteroid D (**1**): colorless needle; [*α*${]}_{D}^{20}$ +0.25 (c 0.32, MeOH); UV (MeOH) λmax (log ε) 203 (2.38) nm; ^1^H and ^13^C NMR data, see Table 1 and Table 2; HRESIMS *m/z* 479.3547 [M + Na]^+^ (calcd for C_30_H_48_O_3_Na, 479.3501).

Penicisteroid E (**2**): amorphous powder; [*α*${]}_{D}^{20}$ +16.6 (*c* 1.09, MeOH); UV (MeOH) *λ*max (log ε) 205 (1.69) nm; ^1^H and ^13^C NMR data, see Table 1 and Table 2; HRESIMS *m/z* 511.3383 [M + Na]^+^ (calcd for C_30_H_48_O_5_Na, 511.3399).

Penicisteroid F (**3**): white powder; [*α*${]}_{D}^{20}$ +19.1 (c 0.31, MeOH); UV (MeOH) λmax (log ε) 205 (2.71) nm; ^1^H and ^13^C NMR data, see Table 1 and Table 2; HRESIMS *m/z* 545.3454 [M + Na]^+^ (calcd for C_30_H_50_O_7_Na, 545.3454).

Penicisteroid G (**4**): white powder; [*α*${]}_{D}^{20}$ +10.5 (c 0.08, MeOH); UV (MeOH) λmax (log ε) 204 (2.20) nm; ^1^H and ^13^C NMR data, see Table 1 and Table 2; HRESIMS *m/z* 529.3518 [M + Na]^+^ (calcd for C_30_H_50_O_6_Na, 529.3505).

Penicisteroid H (**5**): white powder; [*α*${]}_{D}^{20}$ +16.8 (c 0.08, MeOH); UV (MeOH) λmax (log ε) 204 (0.57) nm; ^1^H and ^13^C NMR data, see Table 1 and Table 2; HRESIMS *m/z* 513.3570 [M + Na]^+^ (calcd for C_30_H_50_O_5_Na, 513.3556).

### 3.4. X-Ray Crystallographic Analysis of Compound ***1***

Compound **1** was obtained as a colorless orthorhombic crystal from MeOH. The crystal data were recorded with an Xcalibur Eos Gemini single-crystal diffractometer with Cu Kα radiation (*λ* = 1.54184 Å). Space group P2_1_2_1_2_1_, a = 5.94350(10) Å, b = 11.98440(10) Å, c = 38.3459(4) Å, *α* = *β* = *γ* = 90º, V = 2731.35(6) Å^3^, Z = 4, D_calcd_ = 1.111 mg/cm^3^; crystals size 0.15 × 0.12 × 0.05 mm^3^, *µ* = 0.533 mm^−1^, F (000) = 1008; A total of 18,704 reflections were collected in the range of 2.304 to 67.080°, of which 4726 independent reflections [R (int) = 0.0845] were used for analysis. The final R indices [I > 2sigma(I)] *R*_1_ = 0.0797, w*R*_2_ = 0. 2359. The absolute structure parameter was −0.06 (15). Crystallographic data of **2** have been deposited in the Cambridge Crystallographic Data Center (CCDC), with deposition number 1887656. Copies of the data can be obtained, free of charge, on application to CCDC, 12 Union Road, Cambridge CB21EZ, UK, (fax: +44(0)-1233-336033; email: deposit@ccdc.cam.ac.uk).

### 3.5. Cell Proliferation Assay

Cytotoxic activities of all steroids were conducted on human glioma cell line SHG-44, liver cancer cell lines HepG2, and 7402, non-small cell lung cancer cell line A549, bladder cancer cell line BIU-87, esophageal cancer cell line ECA-109, cervix cancer cell line Hela-S3, pancreatic cancer cell line PANC-1, colon carcinoma cell lines SW620 and HcT116, breast cancer cell lines MCF-7and MB-231 by MTT method as reported previously [32].

### 3.6. Apoptosis Determination by FCM

The levels of cellular superoxide were assessed by using FITC Annexin V Apoptosis Detection Kit (556547, BD Biosciences, San Diego, CA, USA) according to the manufacturer’s instructions. Briefly, A549 cells treated with trypsin and resuspended in Annexin V binding buffer, then the cells were labelled by PI and FITC. At last, fluorescence was measured by flow cytometry using FITC-A (CytoFLEX, Beckman Coulter, Kraemer Boulevard Brea, CA, USA).

### 3.7. Western Blotting

Cell lysates were boiled in sodium dodecyl sulfate (SDS) sample loading buffer, resolved by 10% SDS–polyacrylamide gel electrophoresis (SDS–PAGE) and transferred to nitrocellulose. The membranes were blocked in 5% milk in Tris-buffered saline and Tween 20 (TBST; 10 mM Tris-HCl (pH 8.0), 150 mM NaCl, and 0.05% Tween 20) for 1 h at room temperature. After washing twice with TBST, the membranes were incubated with appropriate primary antibodies (Anti-PARP, 9542, CST, Boston, Massachusetts, USA; Anti-β-actin, 4970S, CST, Boston, Massachusetts, USA; Anti-*RXR**α*, ΔN197, Santa Cruz, 2145 Delaware Ave, CA, USA) in TBST for 1 h and then washed twice, probed with horseradish peroxide-linked anti-immunoglobulin (1:5000 dilution) for 1 h at room temperature. After three washes with TBST, immunoreactive products were visualized using enhanced chemiluminescence reagents and autoradiography.

### 3.8. Cell Cycle Determination by FCM

A549 cells were treated with trypsin then dehydrated with 70% ethyl alcohol overnight. After washing twice by PBS, the cells were labelled by DAPI (1:10000 in PBS, D8417 from Sigma-Aldrich, Saint Louis, MO, USA). At last, fluorescence was measured by flow cytometry using PB450-A (CytoFLEX, Beckman Coulter, Kraemer Boulevard Brea, CA, USA).

### 3.9. Dual-Luciferase Reporter Assay

Cells were transfected with the corresponding plasmids for 24 h and then treated with compounds for 12 h. Cells were lysed and luciferase relative activity was tested by the Dual-Luciferase Reporter Assay System according to the manufacturer’s instructions. Transfection efficiency was normalized to Renilla luciferase activity. Culture medium DMEM containing 0.5% DMSO was regarded as blank control. Celastrol (1 μM), UVI3003 (2 μM, sc-358586, santa cruz, 2145 Delaware Ave, CA, USA), 9-cis (0.1 μM, R4643, Sigma-Aldrich, St. Louis, MO, USA) were used as positive controls.

### 3.10. Protein Expression and Purification

The human *RXRα*-LBD was cloned as an N-terminal histidine-tagged fusion protein in pET15b expression vector and overproduced in the *Escherichia coli* BL21 DE3 strain. Briefly, cells were harvested and sonicated, and the extract was incubated with the His60 Ni Superflow resin.

### 3.11. Fluorescence Quenching Assay

The *RXRα*-LBD protein (1 μM in 3 mL phosphate buffer) was measured with Agilent Technologies Cary Eclipse Fluorescence Spectrophotometer (Agilent Technologies, Malaysia), and the fluorescence spectra were obtained from 300 nm to 500 nm. Compounds were added to the protein. After incubation for 30 s at RT, the incubation buffer was measured with the spectrophotometer. Data were processed and fitted to obtain the binding affinities using Origin (OriginLab, Northampton, MA, USA).

### 3.12. Statistical Analysis

Each experiment was performed three times. The values shown are the mean ± SD. Student’s *t*-test (two-sided) was used to test for significant differences between groups.

## 4. Conclusions

In conclusion, five new and 27 known compounds were isolated from the deep-sea-derived fungus *Penicillium granulatum* MCCC 3A00475. Compounds **2** and **4**–**7** showed moderate activity against the human lung cancer cell line A549. They could induce PARP cleavage and regulate RXRα transcriptional expression.

## Figures and Tables

**Figure 1 marinedrugs-17-00178-f001:**
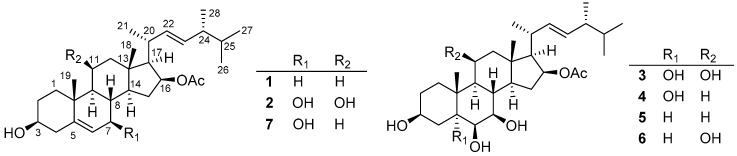
Chemical structures of **1**−**7** isolated from *Penicillium granulatum* MCCC 3A00475.

**Figure 2 marinedrugs-17-00178-f002:**
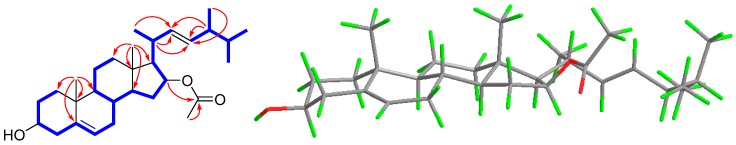
Key COSY (

), HMBC (
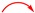
), and NOESY (
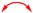
) correlations of **1.**

**Figure 3 marinedrugs-17-00178-f003:**
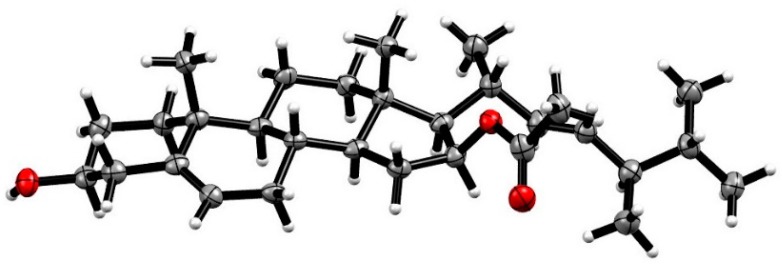
The single X-ray crystallography of **1**.

**Figure 4 marinedrugs-17-00178-f004:**
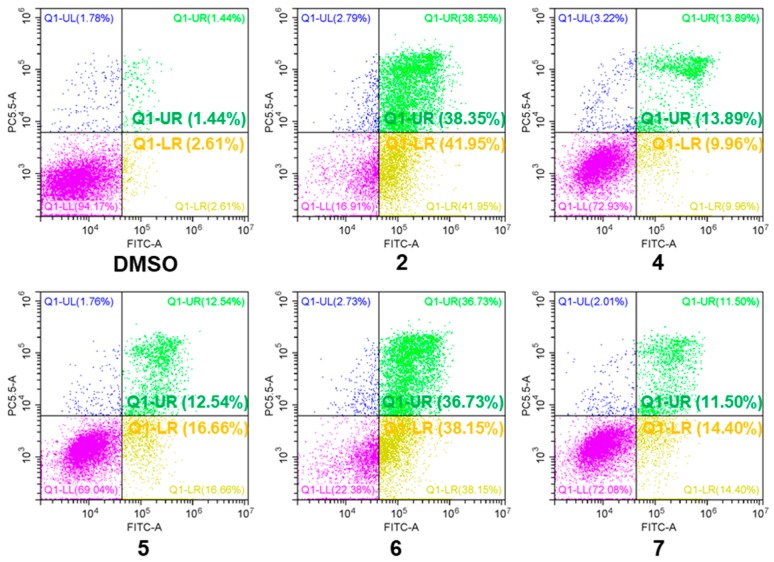
Apoptosis effects of **2** and **4**–**7** in A549 cells. A549 cells were treated with the tested compounds (50 μM) for 9 h. Then they were stained using the Fluorescein Isothiocyanate (FITC) Annexin V Apoptosis Detection Kit and were analyzed by flow cytometry (FCM). Q1-UR: viable apoptotic cells, Q1-LR: non-viable apoptotic cells, Q1-LL: normal cells, Q1-UL: necrotic cells, and the percentage of QI-UR panel indicates the effect of apoptosis.

**Figure 5 marinedrugs-17-00178-f005:**
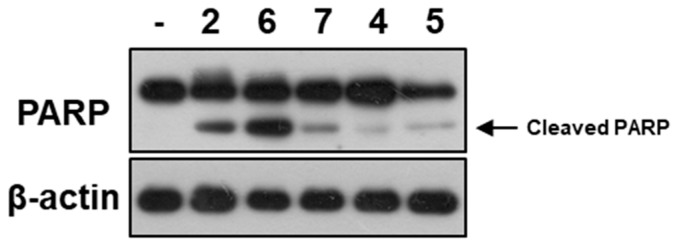
PARP cleavage induced by **2** and **4**–**7**. A549 cells were treated with the tested compounds (50 μM) for 12 h. The cleaved PARP indicated the apoptosis bioactivity.

**Figure 6 marinedrugs-17-00178-f006:**
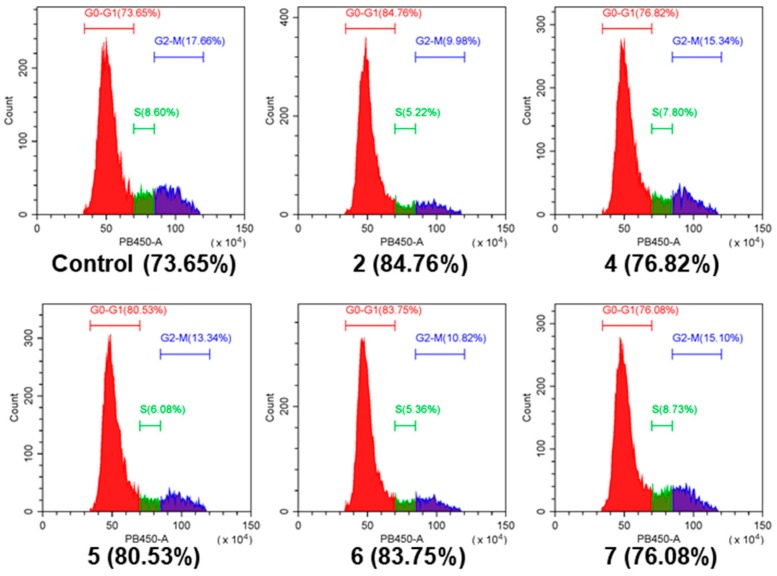
Cell cycle analysis of **2** and **4**–**7**. A549 cells were treated with the tested compounds (25 μM) for 48 h and were stained by 4′6-Diamidino-2-phenylindole dihydrochloride (DAPI). The first red peak indicates cells in the phase G0–G1, the percentages of which are shown next to the group label, and results show the proportion of cells arresting G0-G1 phase increased after treating with compounds **2** and **6**.

**Figure 7 marinedrugs-17-00178-f007:**
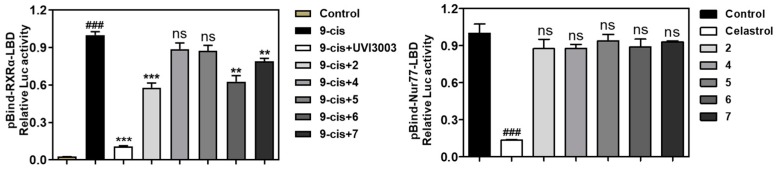
The transcriptional activities of *RXRα* and *Nur77* for **2** and **4**–**7**. (**Left**) Human renal epithelial cells HEK293T transfected dual-luciferase reporter plasmids pBind-*RXRα*-LBD and PG5 were treated with *RXRα* ligand 9-cis-Retinoic acid (9-cis, 0.1 μM), *RXRα* antagonist 3-[4-Hydroxy-3-[5,6,7,8-tetrahydro-5,5,8,8-tetramethyl-3-(pentyloxy)-2-naphthalenyl]phenyl]-2-propenoic acid (UVI3003, 2 μM) and the tested compounds (50 μM) for 18 h, respectively. The luciferase activity of the 9-cis group is normalized to 1. (**Right**) HEK293T cells transfected pBind-*Nur77*-LBD and PG5 plasmids were treated with the tested compounds (50 μM) for 24 h. The luciferase activity of the control group is normalized to 1. ### *p* < 0.001 versus control, ns: no significance, ** *p* < 0.01, *** *p* < 0.001 versus the 9-cis group.

**Figure 8 marinedrugs-17-00178-f008:**
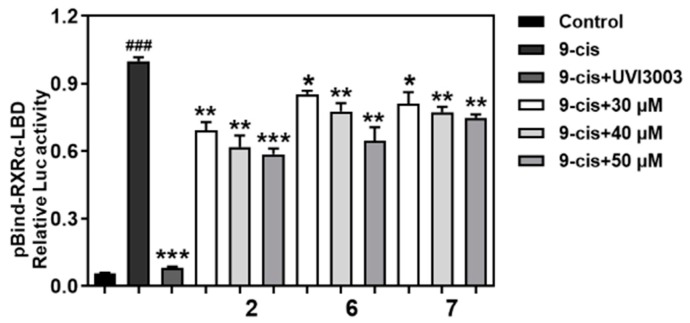
Effects of compounds **2**, **6**, and **7** on the transcriptional activities of *RXRα*. HEK293T cells transfected pBind-*RXRα*-LBD and PG5 plasmids were treated with 9-cis (0.1 μM), UVI3003 (2 μM), and different concentrations of the tested compounds (30, 40, and 50 μM) for 18 h, respectively. The luciferase activity of the 9-cis group is normalized to 1. ### *p* < 0.001 versus control, * *p* < 0.05, ** *p* < 0.01, *** *p* < 0.001 versus the 9-cis group.

**Figure 9 marinedrugs-17-00178-f009:**
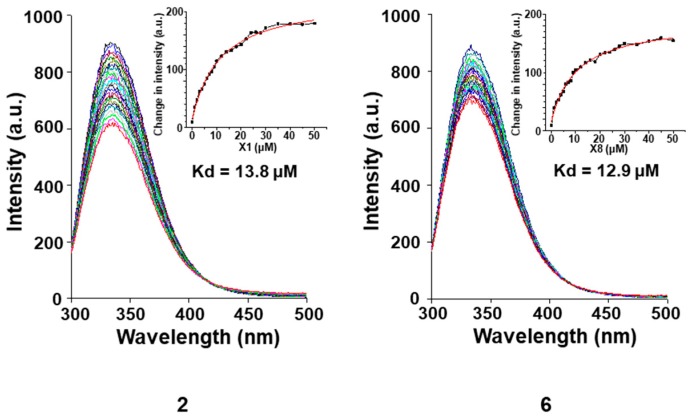
Binding affinity of compounds **2** and **6** to the *RXRα*-LBD. The tested compounds (1–50 μM) were added to the *RXRα*-LBD protein (1 μM in PBS) successively, the fluorescence spectra were obtained from 300 nm to 500 nm using a fluorescence spectrophotometer.

**Figure 10 marinedrugs-17-00178-f010:**
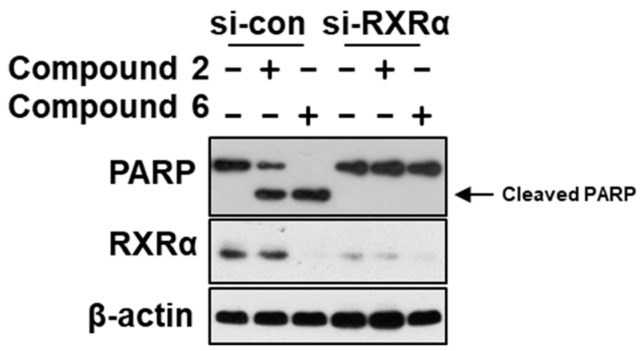
RXRα is required for compounds **2**- and **6**-induced PARP cleavage. A549 cells were transfected with *RXRα* siRNA (SASI_Hs01_00097638, SASI_Hs01_00097639, SASI_Hs01_00097640, Sigma-Aldrich, St. Louis, MO, USA) or control siRNA for 60 h. Then they we1re treated by the tested compound (50 μM) for 12 h, respectively.

**Table 1 marinedrugs-17-00178-t001:** ^1^H NMR data of **1**−**5** at 400 MHz in CD_3_OD (*δ* in ppm, *J* in Hz within parentheses).

No.	1	2	3	4	5
1	1.86 dt (13.2, 3.5); 1.08 dd (13.2, 3.9)	1.98 m; 1.17 m	1.95 m; 1.28 m	1.54 m; 1.31 m	1.63 m; 0.93 m
2	1.77 m; 1.45 m	1.80 m; 1.54 m	1.75 m; 1.56 m	1.75 m; 1.48 m	1.73 m; 1.40 m
3	3.38 m	3.41 m	3.98 m	3.99 m	3.53 m
4	2.20 m	2.27 m	2.17 dd (13.1, 11.4); 1.57 m	2.07 dd (13.0, 11.6); 1.58 m	1.76 m; 1.56 m
5					1.13 m
6	5.32 d (5.0)	5.14 br s	3.37 d (4.5)	3.37 d (4.1)	3.55 m
7	1.93 ddd (12.3, 5.2, 2.2); 1.53 m	3.75 br d (7.3)	3.68 dd (9.8, 4.6)	3.68 dd (10.2, 4.1)	3.16 dd (10.1, 3.6)
8	1.48 m	1.87 m	1.94 m	1.67 m	1.69 m
9	0.96 m	1.10 m	1.54 m	1.41 m	0.73 dt (12.1, 4.6)
11	1.57 m	4.27 dt (3.6, 2.9)	4.12 m	1.42 m	1.58 m; 1.42 m
12	2.04 dt (12.5, 3.3); 1.28 m	2.23 dd (14.2, 2.2); 1.37 dd (14.1, 3.6)	2.20 dd (13.7, 2.6); 1.33 m	2.01 dt (12.7, 3.1); 1.17 m	2.00 m; 1.16 m
14	1.00 m	1.08 m	1.15 m	1.17 m	1.13 m
15	2.30 m, 1.02 m	2.54 m; 1.44 m	2.62 ddd (7.8, 7.6, 6.9); 1.48 m	2.58 m; 1.42 m	2.62 m; 1.43 m
16	5.00 dt (7.6, 3.6)	5.01 dt (7.8, 4.5)	5.00 dt (7.8, 4.5)	5.00 dt (7.8, 4.2)	4.99 dt (7.9, 4.4)
17	1.29 m	1.22 dd (11.1, 7.8)	1.19 dd (10.8, 7.7)	1.21 m	1.22 m
18	0.93 s	1.16 s	1.14 s	0.92 s	0.93 s
19	1.03 s	1.33 s	1.31 s	1.14 s	1.06 s
20	2.53 m	2.54 m	2.52 m	2.52 m	2.53 m
21	1.06 d (6.7)	1.09 d (6.6)	1.08 d (6.8)	1.05 d (6.8)	1.06 d (6.8)
22	5.19 m	5.19 m	5.18 m	5.19 m	5.19 m
23	5.19 m	5.19 m	5.18 m	5.19 m	5.19 m
24	1.76 m	1.75 m	1.78 m	1.76 m	1.75 m
25	1.40 m	1.41 m	1.41 m	1.42 m	1.40 m
26	0.82 d (6.7)	0.82 d (6.7)	0.82 d (6.8)	0.82 d (7.0)	0.82 d (7.0)
27	0.84 d (6.7)	0.84 d (6.7)	0.83 d (6.7)	0.84 d (7.0)	0.84 d (7.0)
28	0.87 d (6.9)	0.87 d (6.7)	0.87 d (6.8)	0.87 d (6.8)	0.87 d (6.9)
2′	1.98 s	1.98 s	1.98 s	1.97 s	1.97 s

**Table 2 marinedrugs-17-00178-t002:** ^13^C NMR spectroscopic data for **1−5** at 100 MHz in CD_3_OD (*δ* in ppm).

No.	1	2	3	4	5
1	38.5 CH_2_	37.1 CH_2_	33.2 CH_2_	33.5 CH_2_	39.7 CH_2_
2	32.3 CH_2_	32.1 CH_2_	31.5 CH_2_	31.7 CH_2_	32.1 CH_2_
3	72.4 CH	71.6 CH	69.0 CH	68.0 CH	72.3 CH
4	43.0 CH_2_	41.8 CH_2_	41.3 CH_2_	41.7 CH_2_	36.1 CH_2_
5	142.3 C	146.0 C	77.9 C	77.8 C	47.4 CH
6	122.2 CH	125.5 CH	78.8 CH	79.3 CH	76.1 CH
7	32.8 CH_2_	74.1 CH	74.2 CH	73.3 CH	77.4 CH
8	32.8 CH	38.2 CH	36.3 CH	39.1 CH	38.6 CH
9	51.7 CH	54.2 CH	48.4 CH	45.4 CH	53.9 CH
10	37.7 C	38.0 C	39.5 C	38.8 C	35.9 C
11	21.9 CH_2_	68.4 CH	68.0 CH	22.2 CH_2_	22.0 CH_2_
12	40.9 CH_2_	49.9 CH_2_	50.2 CH_2_	41.3 CH_2_	41.0 CH_2_
13	43.6 C	43.3 C	44.0 C	44.8 C	44.7 C
14	56.0 CH	57.0 CH	56.6 CH	54.9 CH	54.9 CH
15	36.0 CH_2_	37.5 CH_2_	38.6 CH_2_	38.7 CH_2_	38.8 CH_2_
16	77.1 CH	77.1 CH	77.5 CH	77.6 CH	77.7 CH
17	61.2 CH	61.2 CH	61.2 CH	60.5 CH	60.4 CH
18	13.1 CH_3_	15.4 CH_3_	15.8 CH_3_	13.4 CH_3_	13.3 CH_3_
19	20.6 CH_3_	22.8 CH_3_	20.1 CH_3_	17.6 CH_3_	16.4 CH_3_
20	35.9 CH	35.8 CH	35.9 CH	35.8 CH	35.8 CH
21	21.5 CH_3_	21.5 CH_3_	21.6 CH_3_	21.6 CH_3_	21.5 CH_3_
22	136.8 CH	136.9 CH	137.0 CH	137.0 CH	136.9 CH
23	133.9 CH	133.7 CH	133.7 CH	133.7 CH	133.7 CH
24	44.8 CH	44.7 CH	44.8 CH	44.8 CH	44.7 CH
25	34.3 CH	34.3 CH	34.3 CH	34.3 CH	34.3 CH
26	19.9 CH_3_	20.1 CH_3_	20.6 CH_3_	20.1 CH_3_	20.1 CH_3_
27	20.2 CH_3_	20.5 CH_3_	20.7 CH_3_	20.6 CH_3_	20.5 CH_3_
28	18.7 CH_3_	18.6 CH_3_	18.6 CH_3_	18.6 CH_3_	18.6 CH_3_
1′	172.3 C	170.1 C	172.4 C	172.4 C	172.4 C
2′	21.7 CH_3_	21.7 CH_3_	21.7 CH_3_	21.7 CH_3_	21.6 CH_3_

**Table 3 marinedrugs-17-00178-t003:** The antiproliferative effects (IC_50_) of compounds **1**‒**19** from *Penicillium granulatum* MCCC 3A00475 against 12 tumor cells *^a^*.

No.	SHG-44	HepG2	A549	BIU-87	BEL-7402	ECA-109	Hela-S3	PANC-1
**2**	8.3	NA	5.5	NA	NA	NA	NA	NA
**4**	4.8	6.7	8.0	14.4	8.5	8.3	10.0	5.6
**5**	NA	7.0	4.4	8.5	NA	9.2	7.2	NA
**6**	12.5	6.2	4.5	7.7	8.8	4.1	4.8	4.2
**7**	7.8	NA	4.4	NA	NA	6.6	9.9	7.0
**8**	NA	NA	5.9	NA	NA	15.6	NA	NA
**9**	NA	NA	NA	NA	NA	NA	NA	NA
**12**	NA	NA	7.0	8.7	8.0	NA	NA	6.3
**13**	NA	NA	NA	NA	NA	7.1	9.8	NA
**19**	NA	16.6	NA	NA	7.9	7.5	8.2	NA
**OT** *^b^*	NA	NA	NA	NA	NA	NA	NA	NA

*^a^* Twelve tumor cells included SHG-44 (human glioma cell line), HepG2 (liver hepatocellular cell line), A549 (human non-small cell lung cancer cell line), BIU-87 (human bladder cancer cell line), BEL-7402 (human hepatocellular cell line), ECA-109 (human esophageal cancer cell line), Hela-S3 (Human cervical cancer cell line ), PANC-1 (Human pancreatic cancer cell line), SW620 (human colon cancer cell line), HcT116 (human colon cancer cell line), MCF-7 (human breast cancer cell line), and MB-231 (human breast cancer cell line). Compounds **1**‒**19** did not show positive effect against four tumor cells of SW620, HcT116, MCF-7, and MB-231 (IC_50_ > 20 *µ*M). *^b^* Other compounds, including **1**, **3**, **10**, **11**, and **14**–**18**. NA: No activity was observed (IC_50_ > 20 μM).

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
