# Peer review of "Steroids from the Deep-Sea-Derived Fungus *Penicillium"

_marinedrugs, 2019, doi:10.3390/md17030178_

Round 1
Reviewer 1 Report
Xie and colleagues present an article on the characterization of compounds isolated after cultivating a deep-sea-derived Penicillium species and the involvement of some of those substances on cell proliferation and in RXRα-dependent apoptosis mechanisms. In general the manuscript contains important information and is comprehensibly written.
In my opinion, the Introduction should be modified and expanded a bit in order to also avoid repetition of the same information given in the abstract. Unlike the detailed description of the identification/structural determination part, what seems to be missing from the rest of the manuscript is an actual description of the results of the biological experiments presented in Figures 4-10. For example, Lines 160-163 are the only part of the text where Figures 4, 5 and 6 are mentioned. Thus, in order also to help the readers follow the results, further details on what exactly is depicted in the figures and/or how this correlates with current literature, should be added in the relevant part of the text or in the figure legends.
Other points
Figure 5: Authors may want to consider including a less exposed image of the upper panel, along with the overexposed one. Indicating the cleaved PARP bands with a symbol would also help the readers follow the relevant descriptions and interpretations in the text. A quantification of band intensity may be added if necessary.
Line 204-205: Please indicate the amount of spores used in the inoculum. This information might be useful, as in many other model filamentous fungi the amount of spores in the culture is crucial for triggering secondary metabolite production.
Line 264: please correct “measure” to “measured” and also check the entire manuscript for similar typing errors.
Line 270: Please indicate specifically which antibodies were used for the westerns blot analyses.
Author Response
Response to Reviewer 1#
Xie and colleagues present an article on the characterization of compounds isolated after cultivating a deep-sea-derived Penicillium species and the involvement of some of those substances on cell proliferation and in RXRα-dependent apoptosis mechanisms. In general, the manuscript contains important information and is comprehensibly written.
Thank you very much for your kind comments.
In my opinion, the Introduction should be modified and expanded a bit in order to also avoid repetition of the same information given in the abstract. Unlike the detailed description of the identification/structural determination part, what seems to be missing from the rest of the manuscript is an actual description of the results of the biological experiments presented in Figures 4-10. For example, Lines 160-163 are the only part of the text where Figures 4, 5 and 6 are mentioned. Thus, in order also to help the readers follow the results, further details on what exactly is depicted in the figures and/or how this correlates with current literature, should be added in the relevant part of the text or in the figure legends.
Thanks for the nice suggestions. Accordingly, the Introduction was modified and expanded a little bit in the revised manuscript. Meanwhile, detailed information was provided in the revised legends of Figures 4–10.
Other points
Figure 5: Authors may want to consider including a less exposed image of the upper panel, along with the overexposed one. Indicating the cleaved PARP bands with a symbol would also help the readers follow the relevant descriptions and interpretations in the text. A quantification of band intensity may be added if necessary.
As suggested, the PARP image in Figure 5 was changed to a less exposed one. Meanwhile, an arrow symbol to indicate cleaved PARP was added both in Figures 5 and 10 of the revised manuscript.
Line 204-205: Please indicate the amount of spores used in the inoculum. This information might be useful, as in many other model filamentous fungi the amount of spores in the culture is crucial for triggering secondary metabolite production.
As required, the amounts of spores used in the inoculum was added in the revised manuscript.
Line 264: please correct “measure” to “measured” and also check the entire manuscript for similar typing errors.
Thank you for the careful check. Accordingly, “measure” was revised to “measured” in the revised manuscript.
Line 270: Please indicate specifically which antibodies were used for the westerns blot analyses.
As a matter of fact, three antibodies were used for the westerns blot analyses, including Anti-PARP, Anti-RXRα, and Anti-β-actin. In case of any misunderstanding, the detailed information was added in the revised manuscript.

Reviewer 2 Report
Pleas see the attached file.

Author Response
Response to Reviewer 2#
This article describes the isolation and structure determination of five new steroids isolated form marine fungus Penicillium sp. Also, the authors found that these compounds showed cytotoxicity against some kinds of cells and induced apoptosis by an RXR-dependent mechanism.
In my opinion, this paper contains many issues. I would like to ask the authors to revise them carefully.
Thank you for your critical comments.
The authors mentioned that the antiproliferative activities of these compounds are “potent” or “strong”. But, actually, the IC50 values are not strong. They should say “weak” or “moderate” activities.
Thank you very much for your suggestion. However, as is known to all, the antiproliferative activity could be defined as “potent” or “strong” when the IC50 value of a compound is below 10 μM.
In Figure 1, the stereochemistries at C9 and C14 are lacking.
As required, the stereochemistries at C9 and C14 were added in the revised manuscript.
In the body, I found many mistakes regarding the chemical shift values. For example, in line 73, they mentioned “H-7 (δH 3.75, ddd, J = 12.3, 5.2, 2.2 Hz))”. However, according to Table 1, the chemical shifts of H-7 are1.93 and 1.53. It does not make sense. I would like to ask the authors to revise the whole manuscript carefully.
Thank you for your careful check. The authors are very sorry for the mistake. Accordingly, they were corrected in the revised manuscript.
How did the authors determine the absolute stereochemistry of the compounds? Usually, we cannot determine the absolute stereochemistry by using X-ray analysis. We can determine only relative stereochemistry using X-ray. Please describe the methods for the determination of the absolute structures.
Thank you. It is true that the single X-ray diffraction analysis could be used to determine the relative stereochemistry of the compound when using Mo-Kα radiation. However, it could also be adopted to assign the absolute configuration when using Cu-Kα radiation. In our study, the single X-ray crystallography was introduced with Cu-Kα radiation. The absolute structure parameter was −0.06(15), which unambiguously established the absolute structure of compound 1.

Reviewer 3 Report
In this manuscript, authors have isolated numbers of steroids including five new ergostanes from the liquid culture of the deep-sea-derived fungus Penicillium granulatum MCCC 3A00475 and studied them for their anticancer potency against several cell lines where compounds 2 and 4 -7 show strong activity against A549 cancer cells and they also found that these compounds could regulate the RXRα transcriptional expression.
Authors have elucidated their chemical structure using 1D and 2D NMR such as 1H &13CNMR, NOESY, COSY and HMBC correlations. These NMR spectral characterization of the compounds are impressive.
The study done here is interesting in the direction of finding new drugs for cancer and can catch the attention of readers. In my opinion the introduction needs some improvement and I would suggest adding some more background information in the introduction. Following are my other comments to improve the manuscript.
Figure 5: Western blot seems over exposed. It’s hard to understand if there is equal loading in the lanes. I would suggest redoing the western blot with less exposure or low protein loading.
Figure 6: Change the scale of histogram so that we can distinctly see the difference between the control and the samples.
Figure 10: Quality of western blot can be improved. There is variation in loading and exposure time of the western blot can be tweaked to improve the quality.
Author Response
Response to Reviewer 3#
In this manuscript, authors have isolated numbers of steroids including five new ergostanes from the liquid culture of the deep-sea-derived fungus Penicillium granulatum MCCC 3A00475 and studied them for their anticancer potency against several cell lines where compounds 2 and 4 -7 show strong activity against A549 cancer cells and they also found that these compounds could regulate the RXRα transcriptional expression.
Authors have elucidated their chemical structure using 1D and 2D NMR such as 1H &13C-NMR, NOESY, COSY and HMBC correlations. These NMR spectral characterization of the compounds is impressive.
Thank you for your kind comments.
The study done here is interesting in the direction of finding new drugs for cancer and can catch the attention of readers. In my opinion the introduction needs some improvement and I would suggest adding some more background information in the introduction. Following are my other comments to improve the manuscript.
Thank you for your suggestions. Accordingly, more background information in the Introduction was provided in the revised manuscript.
Figure 5: Western blot seems over exposed. It’s hard to understand if there is equal loading in the lanes. I would suggest redoing the western blot with less exposure or low protein loading.
Thank you very much. As suggested, a less exposed image was provided in Figure 5 of the revised manuscript.
Figure 6: Change the scale of histogram so that we can distinctly see the difference between the control and the samples.
Thank you for your nice suggestion. The authors tried to change the scale of histogram of Figure 6. However, little improvement was observed because most of cells retained in G0-G1 phase in the control and the sample groups. As a solution, the percentage of G0-G1 phase were added next to the corresponding group label to show the difference.
Figure 10: Quality of western blot can be improved. There is variation in loading and exposure time of the western blot can be tweaked to improve the quality.
As suggested, in the revised manuscript, the PARP image in Figure 10 was changed to a more exposed one, and an arrow symbol to indicate cleaved PARP was added.

Round 2
Reviewer 2 Report
This reviewer cannot agree with the authors’ opinion that “However, as is known to all, the antiproliferative activity could be defined as “potent” or “strong” when the IC50 value of a compound is below 10 uM.”
Because compounds that show the IC50 value below 10 uM are defined as just “active” compounds but not “strongly active” compounds. These criteria have been widely accepted by the fields of Natural Products, especially in American Chemical Society.
If the authors want to use “active”, their compounds should have the IC50 values less than 0.1 uM.
Author Response
This reviewer cannot agree with the authors’ opinion that “However, as is known to all, the antiproliferative activity could be defined as “potent” or “strong” when the IC50 value of a compound is below 10 μM.”
Because compounds that show the IC50 value below 10 μM are defined as just “active” compounds but not “strongly active” compounds. These criteria have been widely accepted by the fields of Natural Products, especially in American Chemical Society.
If the authors want to use “active”, their compounds should have the IC50 values less than 0.1 μM.
Thank you very much for the nice suggestion. Accordingly, the antiproliferative activities of compounds 2 and 4–7 were altered to “moderate” in this re-revised manuscript, instead of “potent” or “strong” in the last revised manuscript.